# Comparative Study on Different Remediation Strategies Applied in Petroleum-Contaminated Soils

**DOI:** 10.3390/ijerph17051606

**Published:** 2020-03-02

**Authors:** Jia-Qi Cui, Qing-Sheng He, Ming-Hui Liu, Hong Chen, Ming-Bo Sun, Jian-Ping Wen

**Affiliations:** 1Key Laboratory of Systems Bioengineering (Ministry of Education), Tianjin University, Tianjin 300072, China; cuijiaqi@tju.edu.cn (J.-Q.C.); chenhong98583@163.com (H.C.); 2SynBio Research Platform, Collaborative Innovation Center of Chemical Science and Engineering (Tianjin), School of Chemical Engineering and Technology, Tianjin University, Tianjin 300072, China; 3Sinopec Engineering Group Luoyang R&D Center of Technology, Henan 471000, China; heqs.lpec@sinopec.com (Q.-S.H.); sunmb.lpec@sinopec.com (M.-B.S.); 4CNOOC EnerTech Beijing Research Institute of Engineering and Technology for Safety and Environmental Protection, Tianjin 300457, China

**Keywords:** petroleum-contaminated soils, remediation, bacterial community, biological activity, soil physicochemical properties

## Abstract

Due to the increasing pollution by petroleum hydrocarbons (PHs), it is an important task to develop eco-friendly and highly efficient methods for remediating petroleum-contaminated soils. In this study, bioremediation technology was applied to remediate PHs contaminated soils, and the bacterial community structure and physicochemical characteristics of the soil treated using different bioremediation regimens were analyzed. Compared with the control condition (S0), the PHs removal efficiency of biostimulation (S2) and bioaugmentation (S3) was increased significantly. Combined biostimulation with bioaugmentation (S4) had the highest PHs removal efficiency, up to 60.14 ± 4.12%. Among all the selected remediation strategies (S1–S4, S1: soil moisture content: 25–30%), the bacterial alpha-diversity was higher than in S0. The genera *Acinetobacter*, *Escherichia-Shigella*, *Bacteroides*, *Microbacterium*, and *Parabacteroides* were found to greatly contribute to PHs’ degradation. In the group S4, the PH-degraders and soil enzyme activity were higher than in the other remediation regimens, and these indices gradually decreased in the mid-to-later periods of all remediation tests. Additionally, the abundance of *alkB* and *nah* genes was increased by improving the environmental condition of the microorganism communities. Redundancy analysis (RDA) revealed that the total nitrogen (TN) and total phosphorus (TP) had a positive correlation with total PHs degradation. This study offers insights into the microbial community response to environmental factors during bioremediation, which shows a promoting effect in enhancing the efficiency of PHs remediation.

## 1. Introduction

As important materials for fuel and the chemical industry, petroleum products are widely used in everyday life. With the increasing global petroleum demand, industrial activities surrounding drilling, transportation, and refining have also increased, making it more likely that related petroleum hydrocarbons (PHs) spillage can contaminate the soil and groundwater [1]. PHs are listed as priority environmental pollutants by the US Environmental Protection Agency [2], because their hydrophobicity permits them to reach and accumulate in the food chain [3]. Therefore, the remediation of PH-contaminated soils is presently a significant task.

The current PHs remediation technologies include physical and chemical, as well as biological approaches [4]. Since physical and chemical remediation technologies are often associated with high costs and risk of leading to additional air or water pollution [5], bioremediation stands out as an efficient, cost-effective, and environmentally friendly approach that is increasingly used to remediate PH-contaminated soils [6,7,8]. Among bioremediation strategies, biostimulation and bioaugmentation are the most common approaches to PHs degradation [9,10]. Biostimulation entails increasing the numbers of indigenous microorganisms and enhancing their metabolism of microorganisms through nutrients (nitrogen, phosphorus, potassium, etc.) addition in soil or wastewater [11]. In bioaugmentation, exogenous microbial communities are introduced into PH-contaminated soils with the aim of increasing the diversity and PH-degradation capacity of the microbial community in the soil [12,13]. Both biostimulation and bioaugmentation showed promising results in the remediation of PH-contaminated soils.

In the process of bioremediation, microorganisms play the key role in eliminating PHs from the environment [14]. At the same time, PHs can be utilized as carbon sources to provide energy for the growth of microorganisms, and finally be transformed into nonpolluting substances, or fully mineralized into carbon dioxide and water by microorganisms [15]. In recent years, it has been reported that bacteria, fungi, and even some algae have the ability to degrade PHs [16]. However, the abundance and reproduction of bacteria in the presence of PHs were higher than that of any other microorganisms in the contaminated environment, and bacteria also have multiple metabolic pathways (both aerobic and anaerobic) with which they can degrade PHs [17]. Hence, bacteria have been considered as the primary and most active degraders of petroleum hydrocarbons. At the same time, thousands of different bacterial species were isolated from PH-contaminated soils, and many members of the *Proteobacteria*, *Firmicutes*, and *Actinobacteria* phyla have been proved to have strong capabilities of PHs degradation [18].

Researchers have found that the activity and population of PH-degraders have a profound effect on PHs remediation efficiency [19]. Microbial activity accounts for most of the transformation of contaminants in the soil, so it is essential to establish indices that can be used to predict soil microbial activity. Soil enzymes, such as catalases, lipases, dehydrogenases, and phosphatases, can indirectly reflect the activity of microorganisms and fertility of the soil in the contaminated region, and they can be regarded as an important element for evaluating the chances of contaminated soil bioremediation [20]. At the same time, the physical and chemical characteristics of soil also have a significant effect on the efficiency of PHs remediation through biodegradation. Thus, enzymes and physicochemical soil characteristics can be used to monitor and assess the efficiency of contaminant removal in bioremediation processes. The most probable number (MPN) is a commonly used method for assessing the abundance of microorganisms in contaminated soils [21].

Different bioremediation strategies have been successfully applied to degrade or remove contaminants from polluted soils, but there are few publications on the relationships between microorganisms, biological activity, and soil environmental factors. In this study, five remediation strategies were designed and applied to remediate artificially PH-contaminated soils. The degradation percentages of PHs were quantified, and the bacterial community structure was determined under different remediation strategies through high-throughput sequencing. The soil enzyme activity and the numbers of PH-degraders were examined, with the aim to assess the efficacy and efficiency of different remediation strategies. Additionally, the physicochemical characteristics of the soil were assessed in an attempt to reveal the relationships between the bacterial community, biological activity, and soil environmental factors in the process of PHs’ bioremediation.

## 2. Materials and Methods

### 2.1. Preparation of Artificially PHs-Contaminated Soils

In this work, 50 kg of original soil samples (no PHs contaminated soil) was collected from Tianjin University campus (depth: 10–15 cm). After the soil was sieved, it was polluted for six months (temperature: 20–25 °C; moisture content: 10%–15%) by adding mixed-PHs (10 chemicals). In the initial stage (0 day), the mixed-PHs contents in the polluted soil was 10,317.57 ± 639.00 mg/kg (Appendix A). The soil contained ten hydrocarbons: tridecane, tetradecane, pentadecane, hexadecane, heptadecane, octadecane, eicosane, heneicosane, naphthalene, and phenanthrene. After six months, the microbiological and physicochemical characteristics of the contaminated soil were determined as shown in Table 1.

### 2.2. Microorganisms and Culture Conditions

To study the degradation of PHs, we used the microorganisms *Microbacterium* sp. CICC 10762, *Kocuria marina* CICC 23948, *Micrococcus luteus* CICC 10680, *Kocuria rosea* CGMCC 1.15046, *Staphylococcus capitis* CICC 21722, and *Bacillus odysseyi* DSM 18869. *Microbacterium* sp.10762, *K. marina* 23948, *M. luteus* 10680, and *S. capitis* 21,722 were obtained from China Center of Industrial Culture Collection (CICC). *K. rosea* CGMCC 1.15046 was obtained from China General Microbiological Culture Collection Center (CGMCC). *B. odysseyi* DSM 18,869 was obtained from Deutsche Sammlung von Mikroorganismen und Zellkulturen (DSMZ). All microorganisms were originally isolated from PH-contaminated soil, and these microorganisms were maintained on nutrient agar slants containing (g/L) peptone 10.0, yeast extract 3.0, and NaCl 5.0.

### 2.3. Remediation Strategies for PH Contaminated soils

In this study, five strategies were designed to remediate PH-contaminated soils (Table 2). According to different remediation strategies, samples comprising 5.00 ± 0.26 kg of PHs of the artificial contaminated soil were packed into plastic pots (0.30 m height and internal diameter 0.23 m), and the remediation was continued for 126 days. The plastic pots were placed in a laboratory, where the temperature was kept at 20–25 °C. The soil samples were stirred every five days, to supply oxygen. Each remediation strategy was repeated three times.

### 2.4. Determination of the Content of Alkanes (ALKs) and Polycyclic Aromatic Hydrocarbons (PAHs)

Three samples (per sample comprising 10 g) of the contaminated soil were taken from each pot after 126 days, and the total PHs were extracted according to a published method [22]. The Super Flash Alumina Neutral column (SF 15–24 g, 20.8 mm × 112 mm. Agilent Technologies) was used to separate ALKs and PAHs, using dichloromethane and n-hexane, respectively [22]. The contents of ALKs and PAHs were determined by gas chromatography (GC), on a Bruker 430 GC instrument (Bruker, USA) equipped with a BR-5 capillary column (30.0 m, 0.32 mm id, 0.25 μm df; Bruker). The carrier gas was nitrogen, and the flow ratio was 1:5. The temperature programs for alkanes and PAHs encompassed an initial 5 min at 50 °C, followed by heating at a rate of 15 °C/min to 290 °C, and holding 290 °C for 5 min. The injector temperature was 230 °C, and the injection volume was 1 μL. The eluents were analyzed by using a hydrogen flame ion detector at 280 °C. All samples were measured three times.

### 2.5. Extraction of Bacterial DNA, PCR Amplification, and Sequencing

Soil samples were collected from different remediation pots after 126 days, and the total DNA was extracted by using the Bacterial DNA Kit (Omega, Shanghai, China), according to the manufacturer’s instructions. The DNA content was determined, using a NanoDrop 1000 (Thermo Fisher Scientific, Wilmington, DE, USA), and the DNA was stored at −20 °C. The variable regions (V3-V4) of the 16S rRNA of the bacterial community were amplified by polymerase chain reaction (PCR). The universal primers used for the PCR were 338F (5′-ACTCCTACGGGAGGCAGCAG-3′) and 806R (5′-GGACTACHVGGGTWTCTAAT-3′) [23], which included TruSeq adapter sequences and indices. All libraries were sequenced on an Illumina MiSeq platform (Illumina, San Diego, California, USA).

### 2.6. Quantitative PCR Analysis of the AlkB and Nah Genes

After 126 days, the abundance of alkane monooxygenase (*alkB)* and naphthalene dioxygenase (*nah*) genes in soil samples from different degradation strategies was determined by q-PCR assays. The specific primers for *alkB* and *nah* were as follows: alkB-F (5′-AAYACIGCICAYGARCTIGGICAYAA-3′), alkB-R (5′-GCRTGRTGRTCIGARTGICGYTG-3′), NAH-F (5′-CAAAA(A/G)CACCTGATT(C/T)ATGG-3′), and NAH-R (5′-A(C/T)(A/G)CG(A/G)G(C/G)GACTTCTTTCAA-3′) [24]. The PCR conditions and standard curve construction were in accordance with a previous publication [25]. The amplification efficiency and coefficient (*r^2^*) for *alkB* and *nah* were 97%, as well as 0.998 and 98% and 0.997, respectively.

### 2.7. Detection of Microbial Populations and Enzyme Activity in the Soil

Three samples (per sample comprising 10 g) were collected, via a sampling tube (length: 35 cm, inner diameter 3 cm), from each pot, and the numbers of heterotrophic microorganisms in the contaminated soil were determined every 7 days, using the plate-count method [22] on Luria–Bertani agar comprising (g/L): glucose 10.0, peptone 5.0, yeast extract 2.0, NaCl 5.0, agar powder 20.0, and pH = 7.2. After the agar plates were incubated for 48 h at 30 °C, the numbers of colony-forming units (CFU) were calculated. At the same time, the numbers of TPH-, alkane-, and PAH-degrading microbes in the soil undergoing different degradation strategies were assessed, using the most probable number (MPN) method, every 7 days [22].

The activity of enzymes, including catalase (CAT), lipase (LPS), dehydrogenase (DDA), phosphatase (PPS), and hydrolysis of fluorescein diacetate (FDA), was detected in the soils undergoing different degradation strategies, every 7 days. CAT activity was detected by calculating the H_2_O_2_ disappearance rate [26]. LPS was detected by calculating the amount of butyric acid released from tributyrin [27]. DDA was detected by the reduction of 2,3,5-triphenylterazolium chloride (TTC) to triphenyl formazan. PPS was detected, using p-nitrophenyl phosphate disodium as substrate [28]. FDA hydrolase was detected according to a published method [29].

### 2.8. Determination of Physicochemical Soil Characteristics

The physicochemical characteristics of soils subjected to the different remediation strategies were determined at the beginning (day 0) and the end (day 126) of the experiment. The moisture content and electrical conductivity (EC) were measured, using a TPY-7PC soil analyzer (Zhejiang Top Technology Co.; Ltd., Zhejiang, China). Total carbon (TC), total organic carbon (TOC), and total nitrogen (TN) were determined, using an EA3000 Elemental Analyzer (Euro Vector S.P.A.; Italy). Total phosphorus (TP) was measured, using the molybdenum–antimony anti-spectrophotometric method [30].

### 2.9. Bioinformatic Analysis and Data Availability

One-way analysis of variance (ANOVA) in conjunction with Duncan’s test (*p* < 0.05) was used to examine differences among the remediation strategies and bacterial community structures, respectively. The raw paired-end reads from the Illumina MiSeq platform were merged and assigned, using FLASH and unique barcodes, respectively, and high-quality reads were used for bioinformatics analysis. According to UCLUST algorithm, all the effective reads from each sample were clustered into operational taxonomic units (OTUs) based on 97% sequence similarity [31]. The Shannon, Chao1, and ACE indices were calculated using mothur software. Principal Component Analysis (PCA) was performed by using Quantitative Insights into Microbial Ecology (QIIME) [32]. Linear Discriminant Analysis Effect Size (LEfSe) was utilized to identify biomarkers that can discern among the remediation strategies [33]. The correlation of bacteria was determined by using SparCC (screening criteria: correlation > 0.1 and *p* < 0.05), and the co-expression analysis network was plotted by using Python [34]. Redundancy analysis (RDA) was used to examine the relationships between the bacterial community and physicochemical soil characteristics.

The raw data of bacterial 16S rRNA sequencing were deposited in the European Nucleotide Archive (accession numbers ERS2984634-ERS2984648).

## 3. Results

### 3.1. Degradation Characteristics of PH-Contaminated Soils with Different Remediation Strategies

After 126 days, the residual contents of PHs in soils subjected to different remediation strategies were measured. In S0, the removal efficiency and residual content of total PHs were 8.94 ± 0.41% and 8483.68 ± 359.65 mg/kg, respectively (Figure 1). Compared with S0, the total PH removal efficiency of S1 was increased to 16.31 ± 0.96% (Figure 1). Furthermore, when S2 and S3 were applied to remediate contaminated soils, the removal efficiency of total PHs were significantly increased to 42.53 ± 3.15 and 35.59 ± 2.18%, respectively (Figure 1). Therefore, the remediation efficiency of biostimulation (S2) was higher than that of bioaugmentation (S3). Among all the remediation strategies, the total PHs-removal efficiency of S4 was the highest, reaching up to 60.14 ± 4.12% (Figure 1).

Compared to the initial contaminated soil (ALKs = 7426.76 ± 527.69 mg/kg, PAHs = 1885.69 ± 154.41 mg/kg), the total residual contents of ALKs and PAHs in S0 were respectively decreased to 6685.73 ± 428.51 and 1797.95 ± 125.74 mg/kg (Table 3), but the difference was not significant. By contrast, when S1 was utilized to remediate contaminated soils, the contents of ALKs and PAHs were decreased to 6082.68 ± 518.85 and 1713.92 ± 94.58 mg/kg, respectively (Table 3). Furthermore, the total ALKs and PAHs degradation in S2 was higher than that of S3 (Table 3). Among the different remediation strategies, the residual PHs content of S4 was the lowest, and the contents of ALKs and PAHs were decreased to 2827.54 ± 308.45 and 885.70 ± 96.89 mg/kg, respectively (Table 3). Additionally, under the remediation conditions, the degradation efficiencies of ALKs were higher than those of PAHs in this study (Table 3). Moreover, the degradation ratios of short-chain ALKs (from tridecane to heptadecane) were higher than those of mid-to-long chain ALKs (from octadecane to heneicosane).

### 3.2. Bacterial Community Structure in Soils Subjected to Different Remediation Strategies

#### 3.2.1. Operational Taxonomic Units (OTUs) and Diversity of Bacteria

In this study, all the sequencing coverage of samples were greater than 99.92% (Table 4), indicating that the actual condition of the bacterial community could be reflected by the sequencing results. Under S0, the number of bacterial OTUs was lower than with any of the other remediation strategies (Table 4). Moreover, compared with S0, the number of OTUs increased in the other remediation strategies, particularly in S4, where the number of OTUs reached 650.67 ± 29.94 (Table 4).

Among the different remediation strategies, S0 had the lowest bacterial alpha-diversity indices (ACE, Chao1 and Shannon; Table 4). However, when S2 and S3 were applied to remediate PH-contaminated soils, the bacterial alpha-diversity index between the two was similar (Table 4). In S4, the highest alpha-diversity index of bacterial was obtained, with ACE and Chao1 values greater than 670.00 (Table 4). Additionally, PCA was utilized to compare the differences of beta-diversity, and the results showed that all the remediation strategies were located in different areas (Appendix A), indicating that the bacterial communities under different strategies were significantly different.

#### 3.2.2. Bacterial Community Composition

In the remediation process of PHs-contaminated soils, a total of 37 known and unassigned bacterial phyla were observed (Appendix A). Among them, *Proteobacteria*, *Firmicutes*, *Bacteroidetes*, and *Actinobacteria* were the dominant bacterial phylum in different remediation strategies (Figure 2A).

In S0, *Proteobacteria* was the dominant phylum and accounted for 46.57 ± 0.61% of the total (Figure 2A). The sub-dominant phyla in S0 were *Saccharibacteria* (26.21 ± 0.43%) and *Bacteroidetes* (12.92 ± 0.32%) (Figure 2A). Compared with S0, the dominant phylum of bacteria changed to *Proteobacteria*, *Firmicutes*, and *Actinobacteria* under the other remediation strategies, but *Proteobacteria* is not significantly different in S0–S4 (Figure 2A). *Proteobacteria* were always the dominant phylum and accounted for 39.29 ± 3.46% to 44.30 ± 5.01% from S1 to S4 (Figure 2A). At the same time, the phyla ranked from second to third, according to abundance, in S1 and S2 were *Actinobacteria* (accounting for 15.97 ± 3.17% and 19.30 ± 1.47%) and *Firmicutes* (accounting for 18.49 ± 1.17% and 18.17 ± 3.78%) (Figure 2A). However, the relative abundance of the phylum *Firmicutes* was higher than that of *Actin*obacteria in the remediation strategies S3 and S4 (Figure 2A). Additionally, the relative abundance of the phyla *Saccharibacteria* in S1 to S4 was lower than S0 (Figure 2A). *Alphaproteobacteria*, *Betaproteobacteria*, and *Gammaproteobacteria* were the major classes among the *Proteobacteria.* The total relative abundance of *alpha-+beta-*+ *gamma-proteobacteria* was in the order S0 (45.87 ± 0.49%), S2 (43.03 ± 4.88%), S1 (41.93 ± 3.32%), S3 (40.79 ± 2.09%), and S4 (36.72 ± 3.06%) (Figure 2B). In this study, the dominant classes of bacteria in S0 were *Alphaproteobacteria*, uncultured-*Saccharibacteria*, and *Gammaproteobacteria* (Figure 2B). However, the dominant classes of bacteria shifted under the remediation process of contaminated soils and mainly encompassed *Gammaproteobacteria*, *Alphaproteobacteria*, *Actinobacteria*, and *Clostridia* (Figure 2B). Compared with S0, the relative abundance of *Alphaproteobacteria* and uncultured-*Saccharibacteria* decreased, and those of *Gammaproteobacteria*, *Actinobacteria*, and *Clostridia* increased under the degradation process from S1 to S4 (Figure 2B).

### 3.3. Contribution to and Correlation of Bacteria with the Remediation Process of Contaminated Soils

In the remediation process of contaminated soils, the microbial community plays a key role in the removal of contaminants, and its contribution is affected by environmental conditions. In S0, the phylum *Saccharibacteria* was the taxa that contributed the most to PHs degradation, but in S2, the most abundant taxon shifted to the phylum *Actinobacteria* (Appendix A). Meanwhile, for S1, S3, and S4, the most contribution taxon was not located at the phylum level (Appendix A). In the remediation strategies of S1 and S4, the LDA values of class *Gammaproteobacteria* and *Bacteroidia* were higher than those of other strategies, respectively (Appendix A). Compared with the other remediation strategies, the family *Brucellaceae* was a major contributor in S2 (Appendix A).

When biostimulation and bioaugmentation were applied to remediate the contaminated soils, the genera *Acinetobacter*, *Escherichia-Shigella*, *Bacteroides*, *Microbacterium*, and *Parabacteroides* greatly contributed to PH degradation (Figure 3). According to the co-expression analysis network of the bacterial community, uncultured-*Bacteroidales* S24-7 had a positive correlation with several other bacterial taxa such as *Propionibacterium*, *Lachnospiraceae* NK4A136, *Fusobacterium*, and *Clostridium* 1 (Figure 3). Moreover, *Fusobacterium* and *Propionibacterium* were also positively correlated with the remediation of contaminated soils (Figure 3). By contrast, *Clostridium* 1, *Saccharopolyspora*, *Lachnospiraceae* NK4A136, *Sphingobium*, uncultured-*Bacteroidales* S24-7, *Saccharopolyspora*, uncultured-*Lachnospiraceae Saccharopolyspora*, *Acinetobacter*, and *Sphingobium* were predicated to have a negative correlation with the degradation of PH (Figure 3).

### 3.4. Numbers of PH-Degraders in Soil Samples from Different Remediation Strategies

A total of 5.75 ± 0.37 × 10^7^ heterotrophic microorganisms were counted in the initial PH-contaminated soil (Table 1). When S0 was applied to remediate the soil, the numbers of heterotrophic microorganisms showed a declining trend, and ultimately decreased to 2.95 ± 0.26 × 10^7^ after 18 weeks (Figure 4A). In contrast to S0, the numbers of heterotrophic microorganism in S1 initially slightly increased in the first two weeks and then decreased with a trend similar to S0 (Figure 4A). Moreover, when S2 and S3 were applied to remediate the contaminated soil, the numbers of heterotrophic microorganisms increased during the first three weeks, and growth rate was higher in S3 than S2 (Figure 4A). However, the numbers of heterotrophic microorganisms in S2 and S3 gradually decreased after three weeks, and ultimately their abundance in S3 was higher than in S2 after 18 weeks (Figure 4A). Among the remediation strategies, the abundance of heterotrophic organisms of S4 was highest after 18 weeks, increasing from 5.75 ± 0.37 × 10^7^ to 2.63 ± 0.12 × 10^9^ (Figure 4A).

The abundances of TPH-, alkane-, and PAH-degraders were higher in S1 than in S0 after 18 weeks (Figure 4B–D). Under the conditions of the S2 to S4 strategies, the numbers of TPH-degraders showed a gradual increase in the first five weeks and reached their highest level in S4 after 18 weeks (Figure 4B). The numbers of ALK-degraders in S2 increased in the first three weeks, and then decreased from 9.77 ± 0.20 × 10^4^ in the third week to 4.98 ± 0.17 × 10^4^ in the 18th week (Figure 4C). In the S3 and S4 groups, the numbers of ALK-degraders increased in the first five weeks, and ultimately the numbers in S4 were highest after 18 weeks (Figure 4C). The numbers of PAH-degraders increased in the first three weeks when S2, S3, and S4 were applied to remediate petroleum-hydrocarbon-contaminated soils, and ultimately their abundance was also the highest in S4 among the different remediation strategies (Figure 4D).

### 3.5. Abundance of alkB and Nah Genes in Soil Samples Treated with Different Remediation Strategies

In this work, the abundance of the *alkB* and *nah* genes in the bacterial communities was estimated through q-PCR. As shown in Table 5, the abundance of *alkB* and *nah* showed large variations under different remediation strategies. The *alkB* gene copy number varied from 3.25 ± 0.14 × 10^4^ to 4.20 ± 0.29 × 10^6^ copies per gram dry soil, and the highest abundance of *alkB* gene appeared in S4 (Table 5). Under the different conditions, the abundance of the *nah* gene was lower than that of the *alkB* gene (Table 5). The *nah* gene varied in the range of 1.62 ± 0.05 × 10^3^ to 2.58 ± 0.14 × 10^4^ copies per g of dry soil, and the highest density of *nah* was in S4 (Table 5). Among the different remediation strategies, the lowest abundance of *alkB* and *nah* genes was found in S0 (Table 5).

### 3.6. Enzyme Activities in the Soils Subjected to Different Remediation Strategies

The effects of different remediation strategies on soil enzyme activity was also determined, and the results are shown in Figure 5. In S0, the activities of all tested soil enzymes were lower than in the other remediation strategies (Figure 5). Compared with S0, the activities of soil enzymes were increased when S1 was applied to degrade PHs from the contaminated soil, but the rising trend of enzyme activity was not obvious (Figure 5). When S2 (biostimulation) was applied to remediate the contaminated soil, the activities of CAT, LPS, and FDA were increased in the second week, and the activities of two kinds of soil enzymes (DDA and PPS) were raised in the fifth and third week (Figure 5). Compared with S2, the activities of CAT, LPS, DDA, and FDA were increased in the second week in S3 (bioaugmentation) (Figure 5). Moreover, the activity of PPS gradually decreased after seven days in S3 (Figure 5). Among the different remediation strategies, S4 had the highest activities of all the tested enzymes (Figure 5).

In this work, the activity of soil enzyme in S1–S4 was higher than S0 after 126 days, but the effects of soil enzyme activity were different by remediation strategies (Appendix A). Under the conditions of S0 and S1, the activity of soil enzymes (except LPS) was similar after 126 days (Appendix A). However, compared with S0 and S1, all tested (S2–S4) soil enzyme activity was almost significant increased during different strategies were applied to remediate PHs contaminated soils (Appendix A). When S2 or S3 was utilized to remediate contaminated soils, the enhance effect of CAT activity was not significant, but the others soil enzyme activity were significantly increased (Appendix A). In the remediate process of PHs-contaminated soils, S4 could significantly increase the activity of soil enzyme; hence, the remediation efficiency of PHs in S4 was significantly higher than the others remediation strategies.

### 3.7. Correlation between the Bacterial Community Structure and Environmental Factors

RDA was utilized to investigate the relationship between bacterial community structure and indices of the soil’s physicochemical characteristics. As shown in Figure 6, the samples from different strategies were located in different graph regions, which suggested that all remediation methods were affected by soil environmental factors. At the same time, S0 was significantly different from all the other strategies (S1–S4) (Appendix A). In the samples from the different remediation strategies, the dominant bacterial phyla were mainly *Proteobacteria*, *Firmicutes*, and *Actinobacteria* (Figure 2A). As shown in Appendix A, the *Proteobacteria* had a negative correlation with *Firmicutes* and *Actinobacteria*, which in turn had a positive correlation with the remediation of the contaminated soils. Additionally, the positive correlation numbers of *Firmicutes* were higher than those of all other bacterial phyla (Appendix A). In this study, PHs were regarded as the main pollution, and the results of RDA showed that TN and TP had a positive effect on TPH. Moreover, the TOC, EC, TC, and C/N had a negative correlation with TPH (Appendix A).

In the remediation process, the relative abundance of the phylum *Proteobacteria* in S0 was higher than under any other conditions, and that of S4 was lowest (Appendix A). However, the relative abundances of the phyla *Firmicutes* and *Actinobacteria* were lowest in S0 (Appendix A). Moreover, the relative abundance of the phyla *Firmicutes* and *Actinobacteria* was highest in S2 and S4 (Appendix A), respectively. Interestingly, some soil indices (such as TOC, TC, C/N, and EC) were highest in the S4 group (Appendix A), but the indices of TN and TPH were highest in the S0 strategy (Appendix A). Additionally, the TP index was highest in S2 (Appendix A). The analysis of correlations between the bacterial community structure and physicochemical soil characteristics showed that *Proteobacteria* have a positive correlation with TN, TP, and TPH, while *Firmicutes* and *Actinobacteria* have a negative correlation with these soil indices (Figure 6). By contrast, *Firmicutes* and *Actinobacteria* had a positive correlation with OC, TC, C/N, and EC, while a negative correlation was observed for *Proteobacteria.* Furthermore*,* the positive correlation level of *Firmicutes* was higher than that of *Actinobacteria* (Figure 6).

## 4. Discussion

With the increase of global energy demand, large amounts of PHs have been extracted, transported, and utilized, leading to serious petroleum contamination in soil due to unavoidable spillage and leakage, which makes the problem of cleaning and remediating soils contaminated by PH pollution an important task at present [35]. In this study, biostimulation and bioaugmentation strategies were applied to remediate PH-contaminated soils.

Under control conditions (S0), the degradation efficiency of PHs was lowest among the different remediation strategies, suggesting that the natural conditions and indigenous microorganisms in the contaminated soil had a low capacity for PH degradation [7]. In the remediation process of PH, suitable soil moisture enhanced the microbial activity in the contaminated environment [36]. The soil moisture content of S1 (25.74 ± 1.85%) was significantly higher than that of S0 (9.25 ± 0.73%), and the PH degradation percentages was also significantly increased under the conditions of S1. Since the remediation efficiency of contaminated soils was raised by adding soil moisture, the limiting factors for further increasing the degradation percentage could be the lack of sufficient nutrients and PH-degrading microorganisms in the polluted soil. In recent years, biostimulation and bioaugmentation have been regarded as efficient methods to degrade and remediate contaminated soils. When S2 (biostimulation) and S3 (bioaugmentation) were applied alone, the removal efficiency of PHs reached 42.53 ± 3.15% and 35.59 ± 2.18%, respectively. Additionally, the remediation efficiency of biostimulation was higher than that of bioaugmentation in this study. The nutrient contents of soil were enhanced in S2, which greatly stimulated the growth of microorganisms in the petroleum hydrocarbon polluted soil [37]. This suggested that the lack of nutrients was the main limiting factor for the remediation of the contaminated soil. In our research, the degradation percentage of PHs was significantly increased after the nutrient materials were added to the contaminated soil, indicating that the indigenous microorganisms had a great unused potential to degrade PHs. Furthermore, when exogenous microorganisms are introduced into the new ecosystem, they must first adapt to the environment, and the nutrient materials were also insufficient in the contaminated soil, so that the PH degradation percentage in S3 was lower than in S2 during the early stages of the process. When S4 (combination of biostimulation and bioaugmentation) was applied to remediate the contaminated soil, it resulted in the highest remediation efficiency among the tested degradation strategies. Additionally, PAHs can be toxic and have low bioavailability for microorganisms. Hence, the residual contents of PAHs were higher than those of alkanes after the remediation process.

During the bioremediation process, the microbial community composition and diversity, especially that of bacteria, is regarded as an important element to evaluate the potential for the removal of contaminants [38]. In this work, *Proteobacteria* were a dominant phylum in all the degradation strategies, which was also observed in many different remediation processes of wastes, such as coking wastewater and hypersaline phenol-laden wastewater [39,40]. Therefore, *Proteobacteria* appear to have a stronger potential to adapt to polluted environments. Additionally, many members of the phylum *Firmicutes* with spore-forming ability are able to endure harsh conditions [41]. The bacterial communities had remediation potential for the contaminated soils under the condition of S1, but nutrient content may be a limiting factor for the degradation of PHs. When S2 (biostimulation) was applied to remediate the PH-contaminated soil, adequate N and P were added to provide nutrients for the microorganisms’ reproduction. However, the numbers of OTUs and alpha-indices of S2 were significantly lower than those of S4, which suggested that the diversity of the bacterial community played an important role in promoting the remediation of the contaminated soil. The OTUs and alpha-index of S3 (bioaugmentation) were higher than those of S2, but inadequate nutrients remained a restricting factor for the degradation of petroleum hydrocarbons. Although the petroleum hydrocarbon degradation percentages of S2 and S3 were significantly higher than those of S0 and S1, the disadvantages of applying biostimulation or bioaugmentation alone were obvious. Among the remediation strategies, the OTUs and alpha-diversity indices of S4 were the highest. At the same time, the numbers of TPH-, ALK-, and PAHs-degraders in S4 were also the highest among the different degradation strategies. As a consequence, biostimulation plus bioaugmentation methods had the highest remediation efficiency of PHs. Meanwhile, compared with biostimulation (S2) and bioaugmentation (S3) alone, combining biostimulation with bioaugmentation (S4) exhibited many advantages for remediating PH-contaminated soils, such as providing adequate nutrients, enhancing the diversity of the bacterial community, and increasing the numbers of PH-degraders. In the remediation process of contaminated soils, the microbial communities display different relationships with PH degradation [4]. When indigenous/exogenous bacteria were applied to remediate contaminated soils, several groups were predicted to have a positive or negative correlation with the degradation of PHs in this study, and these results can provide guidance for remediating PH-contaminated soils by introducing exogenous microbial communities. Additionally, by improving the environment of the microbial flora, the abundance of *alkB* and *nah* genes was increased, which suggested that suitable conditions could enhance the reproduction of specifically adapted degrader species that contain these genes. Meanwhile, this also indicated that improvement of the environmental conditions could also increase the biodegradation efficiency of PHs in contaminated soils.

In the remediation process of contaminated soils, the soil enzyme activities can be used as an indicator of efficacy of the microbial community on PH degradation [42]. Consequently, the activities of soil enzymes CAT, LPS, DDA, PPS, and FDA were investigated in this study. CAT, LPS, and DDA are regarded as important indicators of hydrocarbon biodegradation and can be used to evaluate the activities of aerobic microorganisms in the contaminated environment [43,44]. The activity of PPS and FDA can reflect biological parameters such as the number of microorganisms, respiration, and microbial biomass, as well as the overall enzyme activity in the soil [20,29]. Under natural conditions, the lack of nutrients and low microbial populations are the main factors that limit soil enzyme activity, so the PH degradation efficiency of S0 was lowest among the tested remediation conditions. However, under the condition of S4, the soil physicochemical properties and microbial population number were enhanced; hence, the activities of all tested enzymes were highest in S4. Finally, the remediation efficiency of S4 was also highest among different remediation strategies. Interestingly, the soil enzyme activities of S4 increased in the first two to three weeks and then gradually decreased. During the degradation process, the nutrient content and numbers of PH-degraders were adequate in the initial stage, and soil enzyme activities were higher during the initial stage than during later stages.

Soil environmental factors play an important role in the bioremediation of soils contaminated with PHs. The most significant physical and chemical characteristics of soil which can influence the process of bioremediation are moisture, temperature, availability of oxygen, and nutrients (mainly nitrogen, phosphorus, and potassium), as well as the concentration of PHs [45]. The soil quality of the natural soil in S0 (low moisture and low and inadequate nutrients) was significantly different from all other strategies (S1–S4), and the remediation efficiency of S0 was also the lowest. As shown in Figure 6, the TPH had a positive correlation with TN and TP. At the same time, the dominant phylum, *Proteobacteria*, was also positively correlated with TN and TP. Moreover, the contents of nutrient were also added into the soil in the remediation process. Hence, the PH degradation percentages of S2 and S4 were significantly higher than those observed under natural conditions.

## 5. Conclusions

Among all tested remediation strategies (natural conditions, biostimulation or bioaugmentation, and biostimulation plus bioaugmentation), the combination biostimulation with bioaugmentation had the highest remediation efficiency of PHs in this work. In the bioremediation process of contaminated soil, the number of OTUs and diversity of the bacterial community were significantly increased under the effects of biostimulation and bioaugmentation. Under different degradation conditions (S1–S4), the dominant phyla of the bacterial community were *Proteobacteria*, *Firmicutes*, *Bacteroidetes*, and *Actinobacteria*. At the same time, the dominant classes of bacteria shifted under the remediation process of contaminated soils and mainly encompassed *Gammaproteobacteria*, *Alphaproteobacteria*, *Actinobacteria*, and *Clostridia*. Moreover, it was found that the genera *Acinetobacter*, *Escherichia-Shigella*, *Bacteroides*, *Microbacterium*, and *Parabacteroides* greatly contributed to PH degradation. Additionally, the PH-degrader population and soil enzyme activity were enhanced through the addition of nutrients and exogenous microorganisms, whereby the abundance of *alkB* and *nah* genes was also increased. TN and TP were positively correlated with the degradation of PHs in this study.

## Figures and Tables

**Figure 1 ijerph-17-01606-f001:**
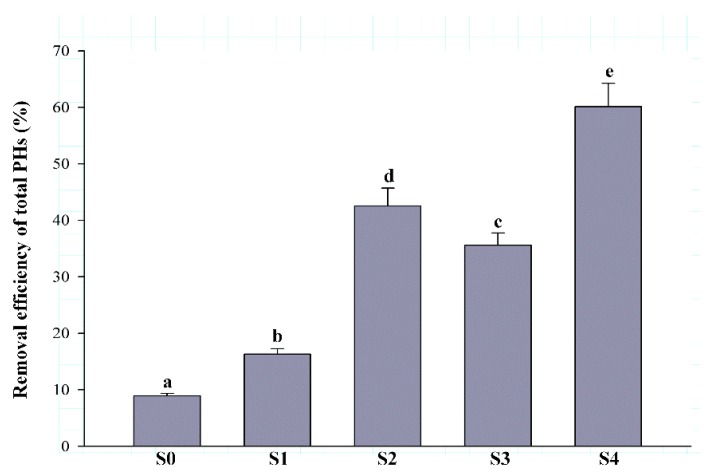
Removal efficiency of total PHs from the contaminated soil. The error bars represent standard deviations, *n* = 3. Lower-case letters indicate significant differences at *p* < 0.05, represents initial contaminated soils.

**Figure 2 ijerph-17-01606-f002:**
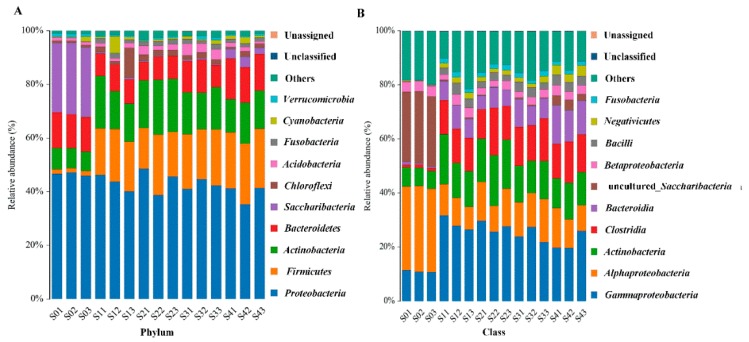
Relative abundance of bacterial at the phylum and class levels. (**A**) represent the phylum of bacterial; (**B**) represent the class of bacterial; S01–S03, S11–S13, S21–S23 S31–S33, and S41–S43 represent the relative abundance of bacteria in S0, S1, S2, S3, and S4, respectively.

**Figure 3 ijerph-17-01606-f003:**
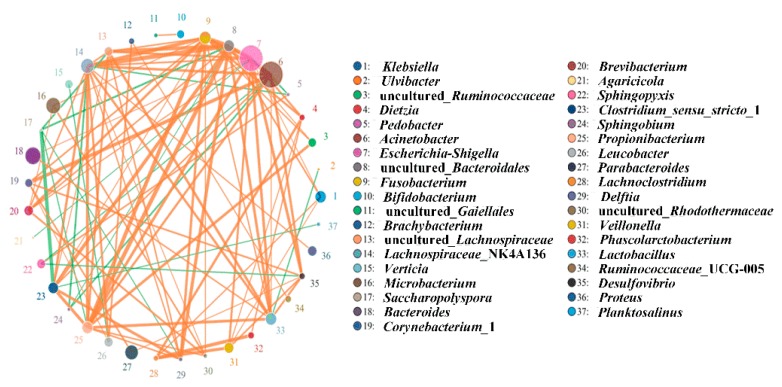
Correlation level of bacteria communities. Circles and sizes represent different genera and relative abundance of bacteria, respectively; lines and thicknesses represent correlation and degree of the bacterial community, respectively; orange and green lines represent positive and negative correlations of the bacterial community, respectively.

**Figure 4 ijerph-17-01606-f004:**
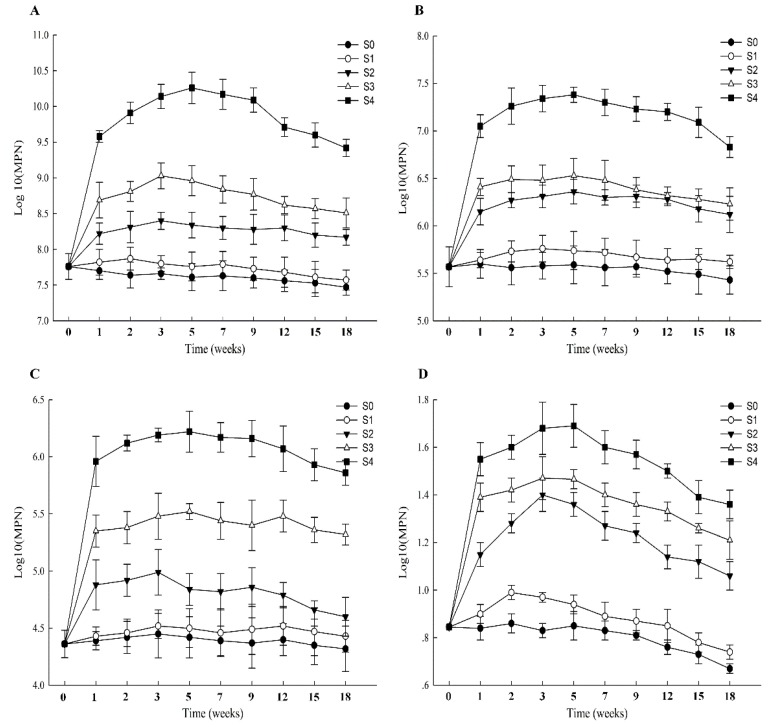
Numbers of PH degraders. Graphs (**A**–**D**) represent the numbers of heterotrophic microorganisms, TPH-degraders, ALK-degraders, and PAH-degraders, respectively.

**Figure 5 ijerph-17-01606-f005:**
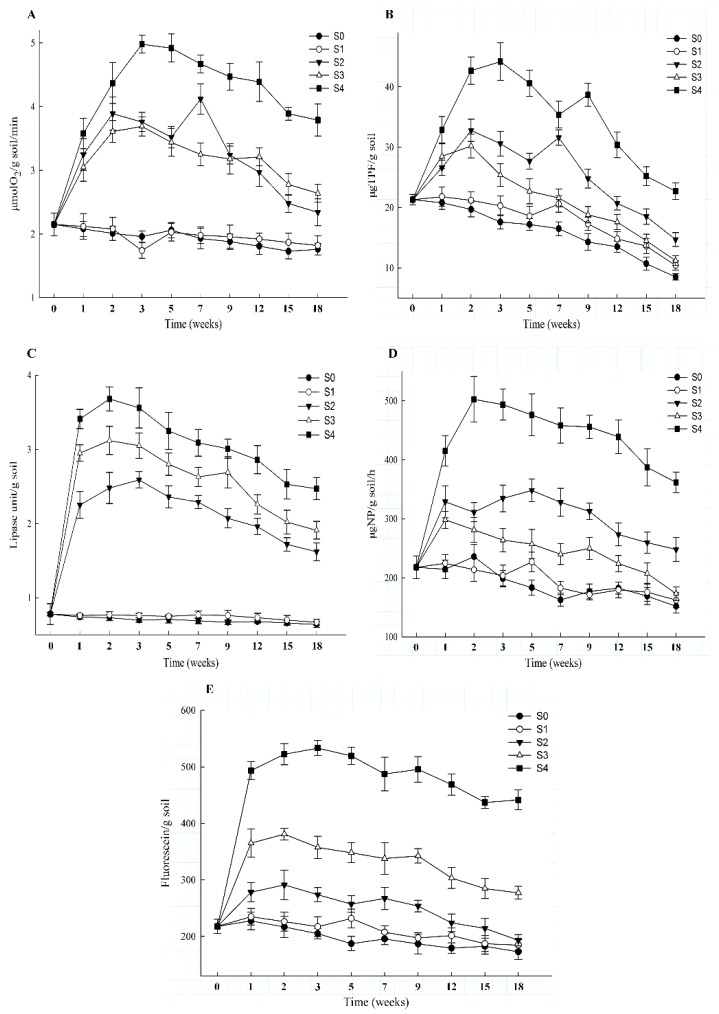
Soil enzyme activities in different remediation strategies. Graphs (**A**–**E**) represent the activities of CAT, LPS, DDA, PPS, and FDA, respectively.

**Figure 6 ijerph-17-01606-f006:**
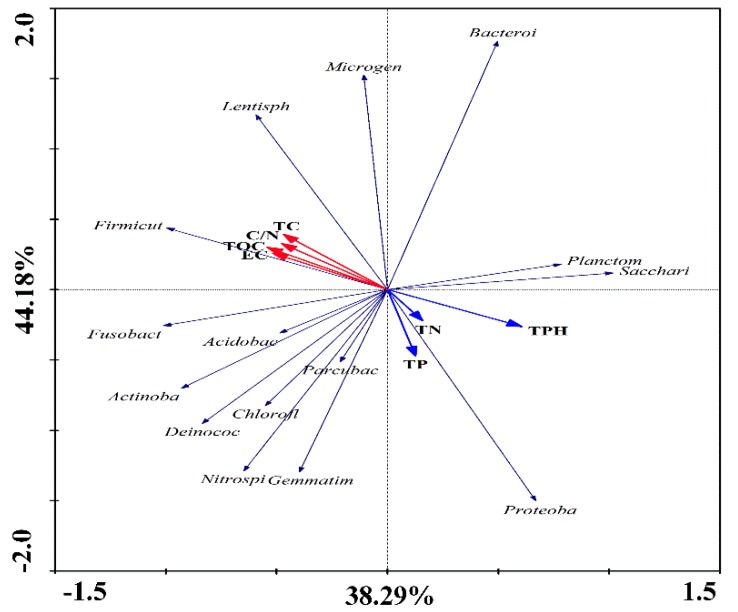
Correlation between bacterial community structure and environmental factors.

**Table 1 ijerph-17-01606-t001:** Microbiological and physicochemical characteristics of PH-contaminated soils.

Petroleum Hydrocarbon	Content (mg/kg)	Soil Characteristics	Values	Microbiological Characteristics	Values
Tridecane	926.32 ± 35.96	Moisture content (%)	12.84 ± 1.36	Heterotrophic bacterial numbers (cell/g)	5.75 ± 0.37 × 10^7^
Tetradecane	912.92 ± 48.75	pH	6.21 ± 0.20	TPH degraders (MPN/g)	3.70 ± 0.21 × 10^5^
Pentadecane	936.57 ± 62.31	Salinity (mg/kg)	73.64 ± 6.26	ALKs degraders (MPN/g)	2.30 ± 0.12 × 10^4^
Hexadecane	915.18 ± 68.55	Electrical conductivity (uS/cm)	134.83 ± 4.66	PAHs degraders (MPN/g)	ND
Heptadecane	925.73 ± 74.21	Total carbon (mg/kg)	462.24 ± 18.96	-	-
Octadecane	936.81 ± 47.98	Total organic carbon (mg/kg)	382.97 ± 12.47	-	-
Eicosane	928.66 ± 53.29	Total nitrogen (mg/kg)	18.65 ± 0.85	-	-
Heneicosane	944.57 ± 67.85	Total phosphorus (mg/kg)	2.57 ± 0.14	-	-
Naphthalene	935.26 ± 74.32	Total potassium (mg/kg)	15.21 ± 0.49	-	-
Phenanthrene	954.39 ± 53.27	C/N ratio	20.59 ± 0.77	-	-
Total	9316.41 ± 625.87	-	-	-	-

ND—not detected; TPH—total petroleum hydrocarbon; ALK—Alkanes; PAHs—polycyclic aromatic hydrocarbons.

**Table 2 ijerph-17-01606-t002:** Different remediation strategies for PH-contaminated soils.

Method	Remediation Strategy
S0	control conditions, no treatment
S1	soil moisture content: 25–30%
S2	0.455 g (NH_4_)_2_SO_4_ and 0.035 g K_2_HPO_4_ were added to contaminated soil; soil moisture content: 25–30%
S3	both 1 × 10^8^ cells/g soil of a microbial consortium (*Microbacterium* sp. CICC 10762, *K. marina* CICC 23948, *M. luteus* CICC 10680, *K. rosea* CGMCC 1.15046, *S. capitis* CICC 21722, and *B. odysseyi* DSM 18869.) were added to contaminated soil; soil moisture content: 25–30%
S4	0.455 g (NH_4_)_2_SO_4_ and 0.035 g K_2_HPO_4_ were added to contaminated soil; a microbial consortium (same to S3) were added to contaminated oil; soil moisture content: 25–30%

**Table 3 ijerph-17-01606-t003:** Residual content of PHs in different remediation strategies.

Method	ALKs (mg/kg)	PAHs (mg/kg)
S0	6685.73 ± 428.71	1797.95 ± 125.74
S1	6082.68 ± 518.85	1713.92 ± 94.58
S2	4125.43 ± 367.22	1228.59 ± 141.52
S3	4690.04 ± 589.11	1310.85 ± 175.48
S4	2827.54 ± 308.45	885.70 ± 96.89

**Table 4 ijerph-17-01606-t004:** OTUs and alpha-diversity of bacteria.

Index	S0	S1	S2	S3	S4
Coverage (%)	99.92 ± 0.02a	99.97 ± 0.01b	99.96 ± 0.02b	99.95 ± 0.02ab	99.95 ± 0.02ab
OTUs	428.00 ± 29.46a	443.33 ± 33.71bc	499.67 ± 54.64bc	491.33 ± 25.01c	650.67 ± 29.94d
ACE	456.23 ± 40.30a	470.48 ± 31.12ab	544.87 ± 50.55bc	542.85 ± 28.80c	672.48 ± 35.55d
Chao1	443.38 ± 48.98a	466.40 ± 45.51ab	552.37 ± 55.79b	536.63 ± 37.14b	671.82 ± 33.95c
Shannon	4.01 ± 0.03a	4.25 ± 0.07a	4.98 ± 0.06b	4.97 ± 0.16b	5.51 ± 0.11b

Coverage represents the sequencing depth; bacteria alpha-diversity was reflected by ACE, Chao1, and Shannon indices; bacterial species richness was reflected by ACE and Chao1 indices; bacterial species evenness was reflected by Shannon. Lower-case letters in the same line indicate significant differences at *p* < 0.05.

**Table 5 ijerph-17-01606-t005:** Abundance of the *alkB* and *nah* genes (copies per g dry soil).

Gene	S0	S1	S2	S3	S4
*alkB*	3.25 ± 0.14 × 10^4^a	4.03 ± 0.29 × 10^4^a	4.29 ± 0.16 × 10^5^b	7.26 ± 0.64 × 10^4^a	4.20 ± 0.29 × 10^6^c
*nah*	1.62 ± 0.05 × 10^3^a	3.55 ± 0.18 × 10^3^b	8.91 ± 0.51 × 10^3^c	4.17 ± 0.33 × 10^3^b	2.58 ± 0.14 × 10^4^d

Lower-case letters in the same line indicate significant difference at *p* < 0.05.

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
