# Peer review of "Comparative Study on Different Remediation Strategies Applied in Petroleum-Contaminated Soils"

_ijerph, 2020, doi:10.3390/ijerph17051606_

Round 1
Reviewer 1 Report
The manuscript “Comparative study on different remediation strategies applied in petroleum‑contaminated soils” tested different strategies for the remediation of petroleum hydrocarbons-contaminated soils, including biostimulation and bioaugmentation. The manuscript is scientifically sound and valid, and will increase the knowledge on the topic. However, some issues need to be addressed before it can be considered suitable for publication in IJERPH. Please refer to my comments listed below.
Section 2.1. The procedure is not clear. The authors stated that no PH were present in the original soil, but then reported the presence of ten species of PH. Please clarify whether the PHs were added or were already present in the soil. Please doublecheck for typos. E.g., L124: “)” Section 2.3.: I would suggest reporting the characteristics of the different tests in a table. Resolution of Fig. 1 is too low. The figures should be enlarged or the fonts’ dimension increased. Similar issues are present in Fig. 2, 3, and 6. Please consider the use of colors for figures 4 and 5. The incipit of the conclusion should be rewritten.Author Response
Dear Editor,
Thank you very much for giving us the chance to revise our manuscript entitled “Comparative study on different remediation strategies applied in petroleum‑contaminated soils” (ijerph-717526).
All the valuable comments provided by reviewers have been taken into full consideration in the revising of the manuscript. All the modifications have been presented by red font and highlighted in yellow color in the revised manuscript. The detailed answers to the comments are summarized separately following this letter. The language of the whole manuscript has also been polished. We hope the revised manuscript could meet the high publication standards of International Journal of Environmental Research and Public Health.
Yours sincerely.
Jianping Wen
Key Laboratory of Systems Bioengineering (Ministry of Education);
SynBio Research Platform, Collaborative Innovation Center of Chemical Science and Engineering (Tianjin);
School of Chemical Engineering and Technology
Tianjin University, Tianjin 300072, China
E-mail: [email protected].
The following is a point-to-point response to the reviewers’ comments on the manuscript entitled “Comparative study on different remediation strategies applied in petroleum‑contaminated soils” (ijerph-717526).
Reviewer #1:
The manuscript “Comparative study on different remediation strategies applied in petroleum‑contaminated soils” tested different strategies for the remediation of petroleum hydrocarbons-contaminated soils, including biostimulation and bioaugmentation. The manuscript is scientifically sound and valid, and will increase the knowledge on the topic. However, some issues need to be addressed before it can be considered suitable for publication in IJERPH. Please refer to my comments listed below.
Question 1:
- Section 2.1. The procedure is not clear. The authors stated that no PH were present in the original soil, but then reported the presence of ten species of PH. Please clarify whether the PHs were added or were already present in the soil.
Response:
Thank you for the reviewer’s valuable suggestions.
According to the reviewer’s comment, we had re-written the Section 2.1 in P2L93-P3L95, please see the revised manuscript.
Question 2:
- Please double check for typos. E.g., L124: “)”.
Response:
Thank you for the reviewer’s valuable suggestions.
We had carefully checked the manuscript again and revised the errors of English writing. Moreover, all the modifications were presented by red font and highlighted in yellow color in the revised manuscript.
Question 3:
- Section 2.3.: I would suggest reporting the characteristics of the different tests in a table.
Response:
Thank you for the reviewer’s valuable suggestions.
We had added a table to reporting the characteristics of the different tests in P4L120, please see the revised manuscript.
Question 4:
- Resolution of Fig. 1 is too low. The figures should be enlarged or the fonts’ dimension increased.
Response:
Thank you for the reviewer’s valuable suggestions.
We had redrawn the Fig.1 in P6L212, please see the revised manuscript.
Question 5:
- Similar issues are present in Fig. 2, 3, and 6.
Response:
Thank you for the reviewer’s valuable suggestions.
We had redrawn the Fig.2, Fig.3 and Fig.6 in P7L260, P8L283, and P13L374, respectively, please see the revised manuscript.
Question 6:
- Please consider the use of colors for figures 4 and 5.
Response:
Thank you for the reviewer’s valuable suggestions.
We were afraid that so many color lines in the figures (4 and 5) to make a confused to watch or read by readers, so we think that a black-white figure is suitable one. Finally, thanks for your advice again.
Question 7:
- The incipit of the conclusion should be rewritten.
.
Response:
Thank you for the reviewer’s valuable suggestions.
According to the reviewer’s comment, we had re-written the incipit of the conclusion in P15L473-P15L475, please see the revised manuscript.
Special thanks to Reviewers’ for your good comments.
Other changes:
We try our best to improve the manuscript and made some changes in the manuscript. These changes will not influence the content and framework of the paper. And all the modifications have been presented by red font and highlighted in yellow color in the revised manuscript.
We appreciate for Editors/Reviewers’ warm work earnestly and hope that the correction will meet with approval.
Once again, thank you very much for your comments and suggestions.
In all, I found the reviewers’ comments are quite helpful, and I revised my paper point-by-point. Thank you and the reviewers again for our help.

Reviewer 2 Report
The work is presented very well. The introduction provided a good backround for the work. The methods are presented clearly. The work addresses the topic in a comprehensive manner.
Note one typo on page 2, line 64 (bacterial). There may be a couple more but overall quality of the writing is very good.
The quality of Fig. 3 needs to be improved
Author Response
Dear Editor,
Thank you very much for giving us the chance to revise our manuscript entitled “Comparative study on different remediation strategies applied in petroleum‑contaminated soils” (ijerph-717526).
All the valuable comments provided by reviewers have been taken into full consideration in the revising of the manuscript. All the modifications have been presented by red font and highlighted in yellow color in the revised manuscript. The detailed answers to the comments are summarized separately following this letter. The language of the whole manuscript has also been polished. We hope the revised manuscript could meet the high publication standards of International Journal of Environmental Research and Public Health.
Yours sincerely.
Jianping Wen
Key Laboratory of Systems Bioengineering (Ministry of Education);
SynBio Research Platform, Collaborative Innovation Center of Chemical Science and Engineering (Tianjin);
School of Chemical Engineering and Technology
Tianjin University, Tianjin 300072, China
E-mail: [email protected].
The following is a point-to-point response to the reviewers’ comments on the manuscript entitled “Comparative study on different remediation strategies applied in petroleum‑contaminated soils” (ijerph-717526).
Reviewer #2:
The work is presented very well. The introduction provided a good background for the work. The methods are presented clearly. The work addresses the topic in a comprehensive manner.
Question 1:
- Note one typo on page 2, line 64 (bacterial). There may be a couple more but overall quality of the writing is very good.
Response:
Thank you for the reviewer’s valuable suggestions.
We had carefully checked the manuscript again and revised the errors of English writing. Moreover, all the modifications were presented by red font and highlighted in yellow color in the revised manuscript.
Question 2:
- The quality of Fig. 3 needs to be improved.
Response:
Thank you for the reviewer’s valuable suggestions.
We had redrawn the Fig.3 in P8L283, please see the revised manuscript.
Special thanks to Reviewers’ for your good comments.
Other changes:
We try our best to improve the manuscript and made some changes in the manuscript. These changes will not influence the content and framework of the paper. And all the modifications have been presented by red font and highlighted in yellow color in the revised manuscript.
We appreciate for Editors/Reviewers’ warm work earnestly and hope that the correction will meet with approval.
Once again, thank you very much for your comments and suggestions.
In all, I found the reviewers’ comments are quite helpful, and I revised my paper point-by-point. Thank you and the reviewers again for our help.

Reviewer 3 Report
Dear authors,
you have a large amount of data and activities to discuss, but in my opinion both the "material and methods" and the "results/discussion" sections require to be greatly improved before the paper could be considered for publication.
In the attached file you can find specific comments and suggestions.

Author Response
Dear Editor,
Thank you very much for giving us the chance to revise our manuscript entitled “Comparative study on different remediation strategies applied in petroleum contaminated soils” (ijerph-717526).
All the valuable comments provided by reviewers have been taken into full consideration in the revising of the manuscript. All the modifications have been presented by red font and highlighted in yellow color in the revised manuscript. The detailed answers to the comments are summarized separately following this letter. The language of the whole manuscript has also been polished. We hope the revised manuscript could meet the high publication standards of International Journal of Environmental Research and Public Health.
Yours sincerely.
Jianping Wen
Key Laboratory of Systems Bioengineering (Ministry of Education);
SynBio Research Platform, Collaborative Innovation Center of Chemical Science and Engineering (Tianjin);
School of Chemical Engineering and Technology
Tianjin University, Tianjin 300072, China
E-mail: [email protected].
The following is a point-to-point response to the reviewers’ comments on the manuscript entitled “Comparative study on different remediation strategies applied in petroleum contaminated soils” (ijerph-717526).
Reviewer #3:
Dear authors,
You have a large amount of data and activities to discuss, but in my opinion both the "material and methods" and the "results/discussion" sections require to be greatly improved before the paper could be considered for publication.
In the attached file you can find specific comments and suggestions.
Question 1:
1. P1Line 23, Is it degradation rate or PHs removal? As indicated 60.14% seems a removal efficiency.
Response:
Thank you for the reviewer’s valuable suggestions.
According to the reviewer’s comment, we had replaced “degradation rate” by “removal efficiency” in P1L23, please see the revised manuscript.
Question 2:
2. P1Line 24, S1 was not previously defined, so it difficult to understand.
Response:
Thank you for the reviewer’s valuable suggestions.
According to the reviewer’s comment, we had added define of SI in P1L24, please see the revised manuscript.
Question 3:
3. P1Line 40, related to?
.
Response:
Thank you for the reviewer’s valuable suggestions.
According to the reviewer’s comment, we had added “related to” in P1L41, please see the revised manuscript.
Question 4:
4. P1Line92, volume
.
Response:
Thank you for the reviewer’s valuable suggestions.
According to the reviewer’s comment, we had added the information of soil collection volume in P1L93, please see the revised manuscript.
Question 5:
5. P1Line93, As working with artificially polluted soil may result in different outcomes than working with real contaminated soil, my suggestion is to improve the description of the procedure used for polluting soil.
Did you use a mixture of PHs? (I suppose a mix of the 10 chemicals here after cited) Did you use a solvent to solubilize the mixture and spread it onto soil?....
Response:
Thank you for the reviewer’s valuable suggestions.
According to the reviewer’s comment, we had re-written the procedure used for polluting soil in in P1L93-P2L95, please see the revised manuscript.
Meanwhile, a mixture of PHs (10 chemicals) was used in this work. Additionally, before the mixture of PHs was spread in onto soil, these chemicals was dissolved in n-hexane.
Question 6:
6. P2Line95, Do you mean just after polluting soil? How many samples did you analyse?
Response:
Thank you for the reviewer’s valuable suggestions.
We had re-written the sentence in P2L95-P2L96, please see the revised manuscript. Meanwhile, in this work, we had collected 3 sites (per 5 g soil) polluted soil to determine the contents of PHs in the initial stage.
Question 7:
7. P2Line99, Is this the uncertainty of the method, or is it the mean value and standard deviation of multiple analyses? How many samples/measurements?
Response:
Thank you for the reviewer’s valuable suggestions.
It is the mean value and standard deviation, and there are 3 samples (per 5 g soil) were determined in this work.
Question 8:
8. P2Line99, check; this table and Fig.4 provide different information.
Response:
Thank you for the reviewer’s valuable suggestions.
The value of Fig.4 was translated (by log10) according to Table 1, so the number information of PHs-degraders was different between the Table 1 and Fig. 4.
Question 9:
9. P2Line100, Do you mean not analysed or below detection limit? In the second case, please provide information of the limit of detection
.
Response:
Thank you for the reviewer’s valuable suggestions.
ND mean not detected.
Question 10:
10. P2Line113, initial soil moisture? 10% as during the soil contamination phase? How did you check and control the moisture level in pots during the tests?
.
Response:
Thank you for the reviewer’s valuable suggestions.
In the natural condition (S0), initial soil moisture (0 day) was 12.84±1.36. However, any factors were not controlled and adjusted in S0 during the soil contamination phase.
During the tests, a TPY-7PC soil analyzer (Zhejiang Top Technology Co.; Ltd, Zhejiang, China) was used to determine the soil moisture each day. When the soil moisture was less than 20%, water was added into the soil and kept the soil moisture in 20%-25%.
Question 11:
11. P2Line122, Did you set up also abiotic controls to check the effects of adsorption onto plastic pots and volatilization?
Response:
Thank you for the reviewer’s valuable suggestions.
This is our mistakes that we did not set up an abiotic control to check the effects of adsorption onto plastic pots and volatilization. In the stage of design experiment, we think that PHs adsorption and volatilization was seldom in this work, so we did not set up an abiotic control. In the future work, we will think about the PHs adsorption and volatilization into the experiment. Finally, thanks for your advice again.
Question 12:
12. P2Line125, Did you monitor O2 with time?
Response:
Thank you for the reviewer’s valuable suggestions.
We must be apologized to you that we did not articulated the experimental process correctly in the section of 2.3, so make you misunderstand that the soil samples were stirred could provide sufficient oxygen. We had corrected error in the 2.3 section in P3L118, please see the revised manuscript.
In this work, we did not monitor O2 with time. Aimed to enhance the PHs degradation, so the soil stirred was used to increase the O2 content in the soil.
Question 13:
13. P2Line125, I'm not sure this one is the right verb.
Response:
Thank you for the reviewer’s valuable suggestions.
This verb (stirred) was cited in a reference, as followed: Wu, M. L.; Li, W.; Dick, W. A.; Ye, X. Q.; Chen, K. L.; Kost, D.; Chen, L. M. Bioremediation of hydrocarbon degradation in a petroleum-contaminated soil and microbial population and activity determination. Chemosphere, 2017a, 169, 124-130.
Question 14:
14. P2Line127, How many sample from each pot?
Response:
Thank you for the reviewer’s valuable suggestions.
According to the reviewer’s comment, I had re-written the sentence in P4L122, please see the revised manuscript.
Question 15:
15. P2Line136, Do you mean each sample has been analysed 3 times?
Response:
Thank you for the reviewer’s valuable suggestions.
In different remediation strategies, three samples of each pot were collected and extracted of total PHs. Meanwhile, each sample was parallel determined three times by GC.
Question 16:
16. P2Line155, Please provide information on soil intermediate sampling (mass of soil, procedure,...); was the sampling performed just after periodic soil mixing for aeration? before only final soil sampling after 126 days was detailed.
Response:
Thank you for the reviewer’s valuable suggestions.
According to the reviewer’s comment, I had re-written the sentence in P4L152- P4L154, please see the revised manuscript.
In this work, the periodic soil mixing and sample collection were 5 days and 7 days, respectively, so the sampling performed was not always after periodic soil mixing for aeration.
Question 17:
17. P2Line155, I suggest providing pertinent references to the bioinformatics analysis (both for the methods and the software tools)
Response:
Thank you for the reviewer’s valuable suggestions.
According to the reviewer’s comment, I had added references in P17L577- P17L584, please see the revised manuscript.
Question 18:
18. P2Line197, Is this the standard deviation based on the analysis of samples from 3 replicate tests?
Response:
Thank you for the reviewer’s valuable suggestions.
It is the standard deviation based on the analysis of samples (S0) from 3 replicate tests.
Question 19:
19. P2Line204, Alk= XXX mg/Kg and PAHs =yyy mg/kg
Response:
Thank you for the reviewer’s valuable suggestions.
According to the reviewer’s comment, I had added the contents of ALKs and PAHs in P6L201-P6L202, please see the revised manuscript.
Question 20:
20. P2Line205, My suggestion is to provide a table summarizing data for the residual contamination in S0-S4; it's difficult to appreciate different removal efficiencies for the different hydrocarbons from Figure 1B.
Response:
Thank you for the reviewer’s valuable suggestions.
According to the reviewer’s comment, I had added Table 3 and re-written the sentence in P6L216 and P6L201-P6L211, respectively, please see the revised manuscript.
Question 21:
21. P2Line216, Do not use repeated colours in Fig 1B (e.g. phenantrene is the same colour of pentadecane). Consider adding furyher figures (eg. for ALKs and PAHs,..).
Response:
Thank you for the reviewer’s valuable suggestions.
According to the reviewer’s comment, I had added Table 3 in P6L216, please see the revised manuscript.
Question 22:
22. P2Line217, n=3?
Response:
Thank you for the reviewer’s valuable suggestions.
According to the reviewer’s comment, I had re-written the sentence in P6L214, please see the revised manuscript.
Question 23:
23. P2Line245, Proteobacteria is not significantly different in S0-S4
Response:
Thank you for the reviewer’s valuable suggestions.
According to the reviewer’s comment, I had re-written the sentence in P7L242-P7L244, please see the revised manuscript.
Question 24:
24. P2Line245, For Saccharibacteria the most relevant difference is between S0 and S1-S4! According to Fig 1A Bacterioidetes seem to remain almost constant in S1-S4. Please check
Response:
Thank you for the reviewer’s valuable suggestions.
According to the reviewer’s comment, I had re-written the sentence in P7L249-P7L250, please see the revised manuscript.
Question 25:
25. P2Line253, What do you mean? I'm sorry but i do not understand...these percentages refer to alpha + beta + gamma proteobacteria?
Response:
Thank you for the reviewer’s valuable suggestions.
According to the reviewer’s comment, I had re-written the sentence in P7L251-P7L253, please see the revised manuscript.
Question 26:
26. P2Line262, Anaerobic? Are you sure periodic mixing was efficient in assuring aerobic conditions? Please discuss.
Response:
Thank you for the reviewer’s valuable suggestions.
We must be apologized to you that we did not articulated the experimental process correctly in the section of 2.3, so make you misunderstand that the soil samples were stirred could provide sufficient oxygen. We had corrected error in the 2.3 section in P3L118, please see the revised manuscript.
In this work, we did not monitor O2 with time. Aimed to enhance the PHs degradation, so the soil stirred was used to increase the O2 content in the soil. Meanwhile, the contents of O2 were not detected, so we did not judge that whether the remediation condition of bacteria community was aerobic or not.
Finally, please forgive our mistakes
Question 27:
27. P2Line285, This paragraph is a bit messy! Comparison should be made at different taxonomic levels (phylum, class/order or family)...do not mix!
Please improve this paragraph
Response:
Thank you for the reviewer’s valuable suggestions.
According to the reviewer’s comment, I had re-written the sentence in P8L265-P8L272, please see the revised manuscript.
Question 28:
28. P2Line285, Figure 3 A and B are to be improved. Both are really hard to read!
Response:
Thank you for the reviewer’s valuable suggestions.
According to the reviewer’s comment, I had redrawn the Fig.3 in P8L283, please see the revised manuscript.
Question 29:
29. P2Line297, growth rate was higher in S3 than S2...
Response:
Thank you for the reviewer’s valuable suggestions.
According to the reviewer’s comment, I had re-written the sentence in P8L295, please see the revised manuscript.
Question 30:
30. P2Line337, Please improve discussion by considering not all the differences are actually significant.
Response:
Thank you for the reviewer’s valuable suggestions.
According to the reviewer’s comment, I had re-written the sentence in P10L336-P10L344, please see the revised manuscript.
Question 31:
31. P2Line374, Several concepts are repeated in this paragraph.
Response:
Thank you for the reviewer’s valuable suggestions.
According to the reviewer’s comment, I had re-written the sentence, please see the revised manuscript.
Question 32:
32. P2Line384, I do not agree. Oxygen diffusion in lower in water than air...Right moisture content (as a rule of thumb about 80% field capacity) improve aerobic biodegradation; excessive moisture creates anaerobic niches.
Response:
Thank you for the reviewer’s valuable suggestions.
According to the reviewer’s comment, I had re-written the sentence in P14L390, please see the revised manuscript.
Question 33:
33. P2Line384, Did you check the residual level of nutrients at the end of the tests?
Response:
Thank you for the reviewer’s valuable suggestions.
In this work, the residual level of nutrients at the end of the tests was determined, as followed:
TC TOC TN TPs TPm
S0 139.63 84.72 13.91 2.29 12.59
S1 195.38 137.88 5.26 1.62 10.37
S2 287.94 347.17 21.48 7.01 29.92
S3 335.73 278.28 4.74 0.92 8.6
S4 627.49 557.93 21.14 6.08 22.77
Question 34:
341. P2Line470, ? check this sentence
Response:
Thank you for the reviewer’s valuable suggestions.
According to the reviewer’s comment, I had re-written the sentence in P15L475-P15L477, please see the revised manuscript.
Special thanks to Reviewers’ for your good comments.
Other changes:
We try our best to improve the manuscript and made some changes in the manuscript. These changes will not influence the content and framework of the paper. And all the modifications have been presented by red font and highlighted in yellow color in the revised manuscript.
We appreciate for Editors/Reviewers’ warm work earnestly and hope that the correction will meet with approval.
Once again, thank you very much for your comments and suggestions.
In all, I found the reviewers’ comments are quite helpful, and I revised my paper point-by-point. Thank you and the reviewers again for our help.

Round 2
Reviewer 3 Report
Dear authors,
you did a great revision of the manuscript.The resolution of figures 3 and 6 still require to be improved; in the attached file you can find few further comments
Best regards
